# Magnetic-Resonance-Imaging-Guided Cryoablation for Solitary-Biopsy-Proven Renal Cell Carcinoma: A Tertiary Cancer Center Experience

**DOI:** 10.3390/cancers16101815

**Published:** 2024-05-10

**Authors:** Mohamed E. Abdelsalam, Nabeel Mecci, Ahmed Awad, Roland L. Bassett, Bruno C. Odisio, Peiman Habibollahi, Thomas Lu, David Irwin, Jose A. Karam, Surena F. Matin, Kamran Ahrar

**Affiliations:** 1Department of Interventional Radiology, The University of Texas MD Anderson Cancer Center, Houston, TX 77030, USA; a.mohamedawad@gmail.com (A.A.); bcodisio@mdanderson.org (B.C.O.); phabibollahi@mdanderson.org (P.H.); tlu3@mdanderson.org (T.L.); dcirwin@mdanderson.org (D.I.); kahrar@mdanderson.org (K.A.); 2Department of Biostatistics, The University of Texas MD Anderson Cancer Center, Houston, TX 77030, USArlbasset@mdanderson.org (R.L.B.); 3Department of Urology, The University of Texas MD Anderson Cancer Center, Houston, TX 77030, USA; jakaram@mdanderson.org (J.A.K.); surmatin@mdanderson.org (S.F.M.)

**Keywords:** MRI, renal ablation, outcomes, RCC

## Abstract

**Simple Summary:**

The aim is to evaluate the long-term efficacy and survival outcome of MRI-guided cryoablation of small renal masses. Our renal ablation database was reviewed. We concluded that MRI-guided cryoablation is a very effective treatment for small kidney masses with long-term disease control.

**Abstract:**

Background: Our purpose is to evaluate the long-term oncologic efficacy and survival rates of MRI-guided cryoablation for patients with biopsy-proven cT1a renal cell carcinoma (RCC). Materials and Methods: We retrospectively reviewed our renal ablation database between January 2007 and June 2021 and only included patients with solitary-biopsy-proven cT1a RCC (≤4 cm) who underwent MRI-guided cryoablation. We excluded patients with genetic syndromes, bilateral RCC, recurrent RCC or benign lesions, those without pathologically proven RCC lesions and patients who underwent radiofrequency ablation or CT-guided cryoablation. For each patient, we collected the following: age, sex, lesion size, right- or left-sided, pathology, ablation zone tumor recurrence, development of new tumor in the kidney other than ablation zone, development of metastatic disease, patient alive or not, date and cause of death. We used the Kaplan and Meier product limit estimator to estimate the survival outcomes. Results: Twenty-nine patients (median age 70 years) met our inclusion criteria. Twenty-nine MRI-guided cryoablation procedures were performed for twenty-nine tumor lesions with a median size of 2.2 cm. A Clavien–Dindo grade III complication developed in one patient (3.4%). Clear cell RCC was the most reported histology (n = 19). The median follow up was 4.5 years. No tumor recurrence or metastatic disease developed in any of the patients. Two patients developed new renal lesions separate from the ablation zone. The 5- and 10-year OS were 72% and 55.6%, respectively. The 5- and 10-year DFS were 90.5% and the 5-year and 10-year LRFS, MFS and CSS were all 100%. Conclusions: MRI-guided cryoablation is a safe treatment with a low complication rate. Long-term follow-up data revealed long-standing oncologic control.

## 1. Introduction

Percutaneous image-guided thermal ablation (TA) has been evolving as an effective strategy in the management of cT1a renal cell carcinoma (RCC) [1,2]. This is highly applicable to non-surgical candidates: elderly patients, those with comorbidities and those with multiple tumors [1,2,3,4]. The excellent periprocedural outcome and reduced complication rate compared to partial nephrectomy (PN) has made TA an attractive treatment option for those patients [5]. In addition, the intermediate- to long-term data have shown oncologic outcomes comparable to those of partial nephrectomy [5,6,7,8,9,10]. Thus, the AUA and NCCN guidelines have included thermal ablation as a treatment option for cT1a RCC [1,11].

Although most percutaneous image-guided cryoablation is performed under computed tomography (CT) imaging guidance, magnetic resonance imaging (MRI) has several advantages, allowing it to be a desirable option for imaging guidance during ablation [12,13,14]. Firstly, the array of MRI pulse sequences available helps to characterize tissue of the target lesion. MRI provides superior contrast resolution, enabling the identification and precise localization of tumors, especially intraparenchymal and endophytic lesions, that may be challenging to visualize with CT or ultrasound. In addition, it provides finer resolution of the tumor boundaries and their relationship to surrounding structures, including blood vessels, without the need for contrast, which can potentially reduce the risk of complications. Moreover, the multiplanar capability is particularly advantageous, enabling precise needle guidance for lesions in difficult locations, such as small upper polar renal lesions, where an oblique approach can help to avoid puncturing the pleural space during the ablation. Also, MRI allows for imaging in any plane, enabling visualization of the entire needle path, even with the double-oblique approach, a feature that recent CT imaging software has begun to incorporate. Furthermore, real-time MRI enables monitoring of the progress of ablation with a better depiction of the ice ball without exposing the patient to ionizing radiation. In comparison to CT imaging, MRI allows for excellent visualization of the ice ball against the retroperitoneal fat. Finally, MRI is a reliable tool that provides immediate assessment of the extension of the ablation zone and ensures complete tumor ablation with an adequate safety margin.

The reported data on MRI-guided renal ablation is limited. A previous study described the technique of MRI-guided cryoablation [12]. Herein, we are evaluating long-term oncologic efficacy and survival rates of MRI-guided percutaneous cryoablation for patients with biopsy-proven cT1a RCC.

## 2. Materials and Methods

We performed a retrospective review of the renal ablation database at our institution. This study was approved by our Institutional Review Board, and a waiver of informed consent was granted. As we started our MRI-guided renal ablation program in 2007, in this study, we included patients who underwent cryoablation between January 2007 and June 2021 for a solitary de novo CT1a RCC (≤4 cm) proven by biopsy. We excluded patients with recurrent RCC, genetic syndromes, bilateral RCC, lesions not histologically proven to be RCC (i.e., benign lesions or lesions with non-diagnostic biopsy) and patients who underwent radiofrequency ablation or cryoablation guided by CT imaging. 

### 2.1. Ablation Procedure 

A detailed description of the procedure for MRI-guided ablation of renal masses was previously published [12]. In our institution, all MRI-guided cryoablation is performed under general anesthesia for renal lesions that are not well depicted on CT, using an MRI-compatible cryoablation system (MRI Seednet Cryoablation System; Boston Scientific, Marlborough, MA USA). Imaging is performed using an 18-channel 1.5 T MRI scanner (MAGNETOM Espree, Siemens Medical System, Erlangen, Germany) [12]. Balanced steady-state free precession (bSSFP) techniques were used for planning and targeting. Real-time imaging with a BEAT IRTTT sequence was used for continuous monitoring during the placement of the probes [12]. Figure 1 shows a case of MRI-guided renal cryoablation for one of the patients included in the current study. At the end of the ablation, a multi-phase contrast-enhanced MRI was performed to confirm complete tumor ablation and identify any immediate complications. Regular follow up with cross-sectional, non-contrast and contrast-enhanced imaging is performed over 24 months and then once a year.

### 2.2. Data Collection

We reviewed each patient’s electronic medical record and documented the demographics, tumor size and side, adjunctive techniques, complications (graded according to the Clavien–Dindo classification system) and pathology (RCC histology, subtype and its Fuhrman grade). We reviewed all available radiological imaging to evaluate for recurrence of the tumor in or outside the ablation zone and for metastatic disease. Also, we recorded if the patients are alive or not at the time of data gathering, date and cause of death and their duration of follow up.

### 2.3. Definitions of Outcomes 

In this study, the primary ablation efficacy represents complete ablation of the tumor in one session and the overall ablation efficacy represents patients who required more than one ablation session to achieve complete tumor ablation. A residual tumor was reported when there was enhancement in the ablation cavity on the first follow-up cross-sectional imaging. Local recurrence describes new contrast enhancement in the ablation zone seen at 6 months or afterwards that was not identified on the prior follow-up imaging. Overall survival (OS) represents the proportion of patients that were alive at last follow up. Local recurrence-free survival (RFS) represents the proportion of patients who had an ablation zone free of any residual/recurrence. Metastasis-free survival (MFS) represents the proportion of patients free of any distant metastatic disease. Disease-free survival (DFS) represents the proportion of patients free of any recurrence of malignancy in the kidneys either inside or outside the ablation zone or distant disease recurrence. Cancer-specific survival (CSS) represents the proportion of patients who did not die from RCC.

### 2.4. Data Analysis and Statistics

Data variables are reported descriptively. We estimated the distribution of OS, RFS, MFS, DFS and CSS using the Kaplan and Meier method. Survival rates were calculated from the date of the procedure to the date of recurrence of the kidney in or outside the ablation zone, distant metastasis or death.

## 3. Results

Twenty-nine patients (eighteen male and eleven female) with a median age of 72 years (range 50–81 years) met the inclusion criteria. 

### 3.1. Procedural Outcomes

A total 29 procedures were performed for 29 biopsy-proven RCC lesions in 29 patients. The median tumor size was 2.2 cm (range 1–4 cm, mean 2.2 cm). The RCC lesions involved the right kidney and left kidney in 21 and 8 patients, respectively. Hydrodissection was performed in 10 patients to displace the bowel/colon (n = 5), spleen (n = 1) or psoas muscle/posterior abdominal wall (n = 4). This adjunctive technique was successful in all cases. The primary ablation efficacy was 100% and none of the patients required a second session to achieve complete tumor ablation. Patient and lesion characteristics are listed in Table 1.

Two patients (6.9%) developed a complication. One patient developed a peri-renal hematoma (Grade I Clavien–Dindo complication) that was managed conservatively by observation and did not require further intervention. The other patient developed pneumothorax (Grade III Clavien–Dindo complication), which required the placement of a temporary chest tube.

### 3.2. Histology Outcomes

This study only included biopsy-proven RCC lesions (n = 29). The most common histological subtype was clear cell RCC (n = 19, 65.5%), followed by papillary RCC (n = 8, 27.5%). One lesion (3.5%) was clear cell papillary RCC and one lesion (3.5%) was chromophobe RCC. Fuhrman grades were reported in 25 lesions: grade 1 in 6 lesions and grade 2 in 19 lesions. The histological characteristics are summarized in Table 2.

### 3.3. Oncologic and Survival Outcomes

All patients were followed up clinically and radiologically with a median follow-up duration of 4.5 years (range 0.9–13.3 years, mean = 5 years). None of the patients had residual disease or local recurrence at the ablation zone during the follow-up period. Two patients (6.9%) developed new lesions: one patient developed a lesion in the ipsilateral kidney separate from the ablation zone and the other patient developed a new lesion in the contralateral kidney. These two new lesions were managed by ablation and active surveillance, respectively. Eleven patients had died at the time of data review, yet none of them died from RCC. The 3-, 5- and 10-year OS were 78%, 72% and 55.6%, respectively. The 3-, 5- and 10-year DFS were 96.6%, 90.5% and 90.5%. The RFS, MFS and CSS were all 100% at 3, 5 and 10 years. Figure 2 shows the Kaplan and Meier graphs of OS and DFS, respectively.

## 4. Discussion

The data reported on the MRI-guided cryoablation of renal tumors and the associated oncologic and survival outcomes are limited. Here, we share our experience at a tertiary cancer center using MRI for guidance during renal cryoablation. In the current study, we outlined the intermediate to long-term oncologic and survival outcomes of MRI-guided cryoablation of histology-proven cT1a RCC. The 5-year OS, RFS, DFS, MFS and CSS were 72%, 100%, 90.5%, 100% and 100%, respectively. The 10-year OS, RFS, DFS, MFS and CSS were 55.6%, 100%, 90.5%, 100% and 100%, respectively. 

We had only two complications (6.9%). This is in concordance with the reported complications in prior studies on MRI-guided renal cryoablation [15,16,17] and matches the complication rates reported following cryoablation in other studies that evaluated the safety and efficacy of cryoablation for RCC [9,18]. The intrinsic high-contrast resolution of MRI enables precise delineation of the tumor borders and its relationship to surrounding structures. This is particularly beneficial for intraparenchymal (endophytic) tumors that may be hard to visualize on non-contrast CT scans. In addition, optimal visualization of the ice ball and hydrodissection in absence of contrast enhance the safety profile of the procedure, diminishing the risk of injury to nearby organs. In our cohort, only one patient (3.4%) had a pneumothorax. There was no injury to the ureteropelvic junction, small bowel, spleen or colon.

Another major advantage of MRI is a lack of ionizing radiation. This advantage addresses a potential concern that was raised with CT-guided renal cryoablation [19]. This advantage is especially applicable when there is a potential for repeat ablation for new lesions in younger patients, patients with syndromes such as Von Hippel Lindau who have multiple lesions and also in younger patients with other non-renal primary malignancies who have an increased likelihood of a cumulative radiation dose due to repeated and long-term cross-sectional imaging follow up.

MRI-guided cryoablation has a few drawbacks. First, the narrow caliber of the bore sometimes presents a challenge when placing multiple probes. Second, the superficial magnetic coil limits the available surface area for probe placement. Third, when multiple probes are in place, the needle artifact interferes with visualization of the tumor. Fourth, MRI-guided procedures generally take a longer time when compared to CT-guided procedures [20].

With regard to survival outcomes, we reported a 5-year OS of 72%, which is lower compared to the data of OS reported for MRI-guided cryoablation [15,20]. This lower OS rate is mostly secondary to the nature of our patient population in a tertiary cancer center as published in a prior report [21]. In the current study, 18 of the 29 (62%) patients had a concurrent non-renal malignancy. As reported in the past, the presence of another non-renal malignancy had an impact on OS compared to patients with only RCC [21]. 

Here, we have presented promising 5-year survival rates, specifically, our 5-year estimate of LRFS (100%). Our survival rates align closely with those reported by other investigators. In a recent single-arm retrospective study, Cazzato et al. [15] presented their experience with MRI-guided cryoablation for renal tumors over a period of 10 years. The authors reported the oncologic and survival outcomes in 143 patients. They reported a 5-year secondary RFS of 91.1%, MFS of 91.5%, DFS of 75.1% and CSS of 98.2%. Likewise, Bhagavatula et al. [20] evaluated the intermediate- to long-term outcomes of image-guided percutaneous cryoablation of cT1 RCC. The authors reported a 5-year RFS of 94%, DFS of 92% and CSS of 100% in the subgroup of patients who underwent MRI-guided cryoablation (n = 152). This is also in concordance with survival data of cryoablation in the literature, where most of these data are for patients who underwent CT-guided cryoablation [9,10].In a prospective study by Georgiades et al. [9,10], the safety, effectiveness and outcomes of percutaneous cryoablation for biopsy-proven RCC were investigated. This study involved 134 patients with T1a renal tumors. The 5-year efficacy rate was 97%. The reported rates of MFS and CSS were both 100%. In a more recent retrospective single-institution study, Breen et al. [9,10] reported their experience with the cryoablation of 268 biopsy-proven RCC cases. The treatment efficacy after one ablation session was 95.6% for all treated lesions, increasing to 98.1% with subsequent sessions in this group of patients. Over a 5-year period, six patients experienced local recurrence, with corresponding rates of recurrence-free survival (RFS) and metastasis-free survival (MFS) at 93.9% and 94.4%, respectively. Duus et al. [22] documented their experience involving 122 patients who underwent cryoablation for biopsy-proven T1 RCC tumors, totaling 128 cases, with 117 classified as T1a RCC. The 3-year disease-free survival rate for the T1a RCC subgroup was 95%.

We acknowledge the limitations of our study. First, it is retrospective in nature with an inherent bias in patient selection. Second, the number of patients included in the study is small. Third, it is a single-arm study with no control group of patients who underwent CT-guided cryoablation. Fourth, the widespread application of the results of this study is limited given the limited availability of MRI interventional suites. However, our study adds to the growing body of evidence on the safety and oncologic effectiveness of MRI-guided cryoablation.

## 5. Conclusions

MRI-guided cryoablation is a safe and highly efficacious modality for treatment of small RCCs. Intermediate- to long-term follow-up data reveal long-standing favorable oncologic and survival outcomes with low complication rates. 

## Figures and Tables

**Figure 1 cancers-16-01815-f001:**
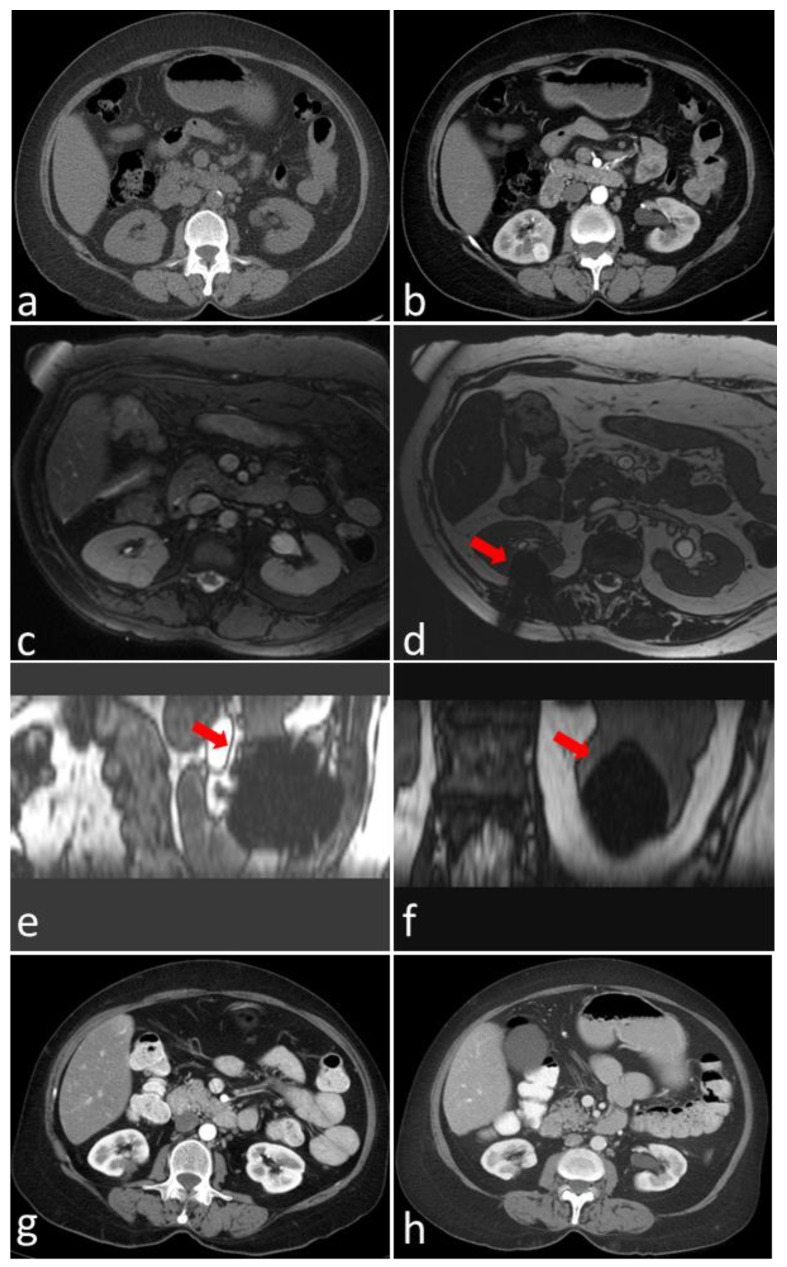
Non-contrast CT (**a**) and contrast-enhanced CT (**b**) for a patient with biopsy-proven renal cell carcinoma of the right kidney. The tumor is isodense on non-contrast CT; thus, the decision was made to perform MRI-guided renal ablation. (**c**–**f**) MRI-guided renal cryoablation procedure. (**c**) Axial true fast imaging with steady-state free precession (TRUFI) with fat suppression showing the 1.5 cm renal mass. (**d**) Axial, (**e**) sagittal and (**f**) coronal TRUFI showing the hypointense ice ball (arrow) during cryoablation. (**g**,**h**) Follow-up contrast-enhanced CT images at 6 months (**g**) and 10 years (**h**) after the procedure.

**Figure 2 cancers-16-01815-f002:**
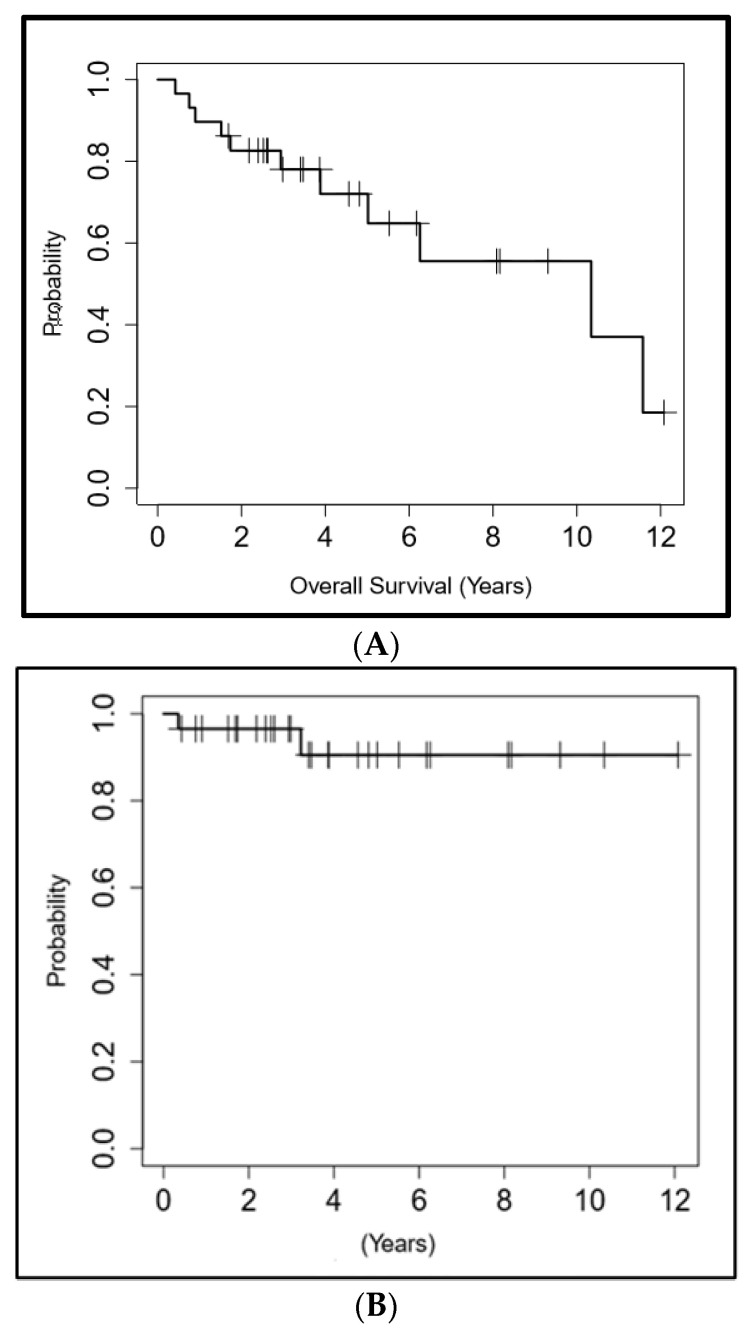
Kaplan–Meier survival curves. (**A**) Overall survival; (**B**) Disease free survival.

**Table 1 cancers-16-01815-t001:** Patient and tumor characteristics.

Characteristic	n (%)
Age, years	
Mean (SD)	72.6 (10.2)
Range (median)	50–81 (70)
Gender	
Female	11 (37.9)
Male	18 (62.1)
Size of lesion, cm	
Mean (SD)	2.2 (0.65)
Range (median)	1.0–4.0 (2.2)
Laterality	
Right	21 (72.4)
Left	8 (27.6)
Complications	
N	27 (93.1)
Y	2 (6.9)
Clavien–Dindo complication grade	
Grade I	1 (3.4)
Grade II	1 (3.4)

**Table 2 cancers-16-01815-t002:** Biopsy histology outcome.

Characteristic	n (%)
Histological type	
Clear cell	19 (65.5)
Papillary	8 (26.5)
Chromophobe	1 (3.5)
Clear cell papillary	1 (3.5)
Fuhrman nuclear grade	
1	6 (20.5)
2	19 (65.5)
NR*	4 (14)

RCC, renal cell carcinoma; NR*, not reported.

## Data Availability

The datasets generated during and/or analyzed during the current study are available from the corresponding author upon reasonable request.

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
