# Peer review of "Magnetic-Resonance-Imaging-Guided Cryoablation for Solitary-Biopsy-Proven Renal Cell Carcinoma: A Tertiary Cancer Center Experience"

_cancers, 2024, doi:10.3390/cancers16101815_

Round 1

Reviewer 1 Report

Comments and Suggestions for Authors

Paper presents concise description of the outcome of MRI guided cryoablation for renal cell carcinoma cases follow-up, assessed during period from 2007 to 2021, and available in author's institution database.  A set of parameters defining long term oncologic efficacy and patient's survival rates were assessed using Kaplan-Meier estimator. 

This represents valuable contribution to the still limited literature, concerning application of the MRI guided cryoablation.

My main concern is the lack of more extended comparative discussion of author's results with work of others, based on both MRI- and CT- guided cryo-biopsy. Only brief mention, suggesting similarity of the outcomes is presented in the discussion. I strongly recommend to elaborate it in more details.

Minor editorial comment: please consider to place figure 1 in one graph, with consistent axis description.

Author Response

Comments and Suggestions for Authors

Paper presents concise description of the outcome of MRI guided cryoablation for renal cell carcinoma cases follow-up, assessed during period from 2007 to 2021, and available in author's institution database.  A set of parameters defining long term oncologic efficacy and patient's survival rates were assessed using Kaplan-Meier estimator. 

This represents valuable contribution to the still limited literature, concerning application of the MRI guided cryoablation.

Author: Thank you so much for your comment.

My main concern is the lack of more extended comparative discussion of author's results with work of others, based on both MRI- and CT- guided cryo-biopsy. Only brief mention, suggesting similarity of the outcomes is presented in the discussion. I strongly recommend elaborating it in more details.

Author: Thank you so much for your comment. Changes have been made to the manuscript.

Minor editorial comment: please consider placing figure 1 in one graph, with consistent axis description.

Author: Thank you so much for your comment. Changes have been made to the manuscript.

Reviewer 2 Report

Comments and Suggestions for Authors

The current manuscript aimed to evaluate the long-term oncologic efficacy and survival rates of MRI-guided cryoablation for patients with biopsy-proven cT1a renal cell carcinoma (RCC). The retrospective study relied on 29 patients over 15 years. Despite the knowledge of findings, several limitations should be acknowledged in the data interpretation.

First, the sample is too scant to generalize the results as the authors did in the conclusion section "MRI-guided PCA is a safe and highly efficacious modality for treatment of small RCCs. Intermediate to long-term follow-up data reveal long-standing favorable oncologic and survival outcomes with low complication rates." Nonetheless, applying Kaplan Meier plots to 29 patients is useless if there is no possibility of weighting the results according to a multivariable model, such as Cox or logistic regression. A longer follow-up should be used to increase the potentiality of the data presented and thus the sample size. 

The authors should be consistent with the descriptive nature of the study. With a such small sample size, they can just describe the overall complications and the number of recurrences. Unfortunately, an adjustment for tumor and patients' characteristics is needed but the current study does not allow it. This inherent limitation reduces the applicability of the study to the general population. The sample is not representative at all.  Other investigators relied on a wider cohort of patients (PMID 32897930, 37225969).

Moreover, is there any information about the PADUA or RENAL score of these lesions? It would be interesting to know the complexity of the procedure in which this novel technique was accounted for. 

MINOR POINTS

The abbreviation PCA for MRI-guided cryoablation (PCA) should be abandoned. PCA is prostate cancer, universally. Please consult the MESH criteria for reporting abbreviations. 

Author Response

Comments and Suggestions for Authors

The current manuscript aimed to evaluate the long-term oncologic efficacy and survival rates of MRI-guided cryoablation for patients with biopsy-proven cT1a renal cell carcinoma (RCC). The retrospective study relied on 29 patients over 15 years. Despite the knowledge of findings, several limitations should be acknowledged in the data interpretation.

Author: Thank you so much for your comment.

First, the sample is too scant to generalize the results as the authors did in the conclusion section "MRI-guided PCA is a safe and highly efficacious modality for treatment of small RCCs. Intermediate to long-term follow-up data reveal long-standing favorable oncologic and survival outcomes with low complication rates." Nonetheless, applying Kaplan Meier plots to 29 patients is useless if there is no possibility of weighting the results according to a multivariable model, such as Cox or logistic regression. A longer follow-up should be used to increase the potentiality of the data presented and thus the sample size. 

Author: Thank you so much for your comment. We will continue to follow up and monitor those patients and hopefully add more cases in the future

The authors should be consistent with the descriptive nature of the study. With a such small sample size, they can just describe the overall complications and the number of recurrences. Unfortunately, an adjustment for tumor and patients' characteristics is needed but the current study does not allow it. This inherent limitation reduces the applicability of the study to the general population. The sample is not representative at all.  Other investigators relied on a wider cohort of patients (PMID 32897930, 37225969).

Author: Thank you for your comment. As noted in the manuscript, the study's sample size represents one of its limitations. Nonetheless, our findings align with those reported in larger patient cohorts, thereby reinforcing the robustness of our results, and facilitating reproducibility. Changes have been to the manuscript to include additional studies that delve into the outcomes of ablation procedures.

Moreover, is there any information about the PADUA or RENAL score of these lesions? It would be interesting to know the complexity of the procedure in which this novel technique was accounted for. 

 Author: Thank you so much for your comment. We didn’t collect data for PADUA or RENAL score of these lesions as multiple studies have shown to benefit for these scoring systems for ablation as they are for surgery.

MINOR POINTS

The abbreviation PCA for MRI-guided cryoablation (PCA) should be abandoned. PCA is prostate cancer, universally. Please consult the MESH criteria for reporting abbreviations. 

Author: Thank you so much for your comment. Changes have been made to the manuscript.

Round 2

Reviewer 2 Report

Comments and Suggestions for Authors

The authors should be congratulated for their endeavors. The comments were addressed properly and the manuscript now reads very well. All the statements were changed into mild sentences, resulting in consistency with t the study limitations. However, the number at risk in the Kaplan-Meier curve should be included.